# Highly Efficient *n*-Type Doping of Graphene by Vacuum Annealed Amine-Rich Macromolecules

**DOI:** 10.3390/ma13092166

**Published:** 2020-05-08

**Authors:** Young-Min Seo, Wonseok Jang, Taejun Gu, Dongmok Whang

**Affiliations:** 1School of Advanced Materials Science & Engineering, Sungkyunkwan University, Suwon 16419, Korea; black4504@skku.edu (Y.-M.S.); wid0129@skku.edu (W.J.); gtj1008@skku.edu (T.G.); 2SKKU Advanced Institute of Nanotechnology (SAINT), Sungkyunkwan University, Suwon 16419, Korea

**Keywords:** flexible transparent conducting electrodes, graphene, doping, charge transfer, sheet resistance, transmittance, figure of merit, stability

## Abstract

Flexible transparent conducting electrodes (FTCE) are an essential component of next-generation flexible optoelectronic devices. Graphene is expected to be a promising material for the FTCE, because of its high transparency, large charge carrier mobilities, and outstanding chemical and mechanical stability. However, the electrical conductivity of graphene is still not good enough to be used as the electrode of an FTCE, which hinders its practical application. In this study, graphene was heavily *n*-type doped while maintaining high transmittance by adsorbing amine-rich macromolecules to graphene. The *n*-type charge-transfer doping of graphene was maximized by increasing the density of free amine in the macromolecule through a vacuum annealing process. The graphene adsorbed with the *n*-type dopants was stacked twice, resulting in a graphene FTCE with a sheet resistance of 38 ohm/sq and optical transmittance of 94.1%. The figure of merit (FoM) of the graphene electrode is as high as 158, which is significantly higher than the minimum standard for commercially available transparent electrodes (FoM = 35) as well as graphene electrodes doped with previously reported chemical doping methods. Furthermore, the *n*-doped graphene electrodes not only show outstanding flexibility but also maintain the doping effect even in high temperature (500 K) and high vacuum (~10^−6^ torr) conditions. These results show that the graphene doping proposed in this study is a promising approach for graphene-based next-generation FTCEs.

## 1. Introduction

In recent years, flexible transparent conducting electrodes (FTCEs) have attracted tremendous interest due to their pivotal role in foldable, flexible and wearable optoelectronic devices [1]. For the past several decades, indium tin oxide (ITO) has been commonly used in the application of transparent conducting electrodes due to its low sheet resistance and high transmittance at the visible region (10–50 ohm∙cm^−2^ at 90% transmittance) [2]. However, the brittle nature of ITO limits the scope of its practical applications for flexible electronic devices. To replace the conventional ITO electrode in flexible devices, various studies on next-generation FTCEs such as conducting polymer [3], metallic nanowire [4], metal grid [5] and carbon-based materials [6] have been conducted. In particular, graphene is a promising candidate for FTCE material due to its remarkable mobility, transparency and flexibility [7,8]. In addition, graphene has bipolar characteristics, so it has the advantage of controlling its characteristics with *n*-type and *p*-type doping. However, since graphene itself does not have sufficient charge carriers, the resistance is relatively high for use as an electrode [9,10,11]. Therefore, further processing is needed to increase the carrier density of graphene.

Doping is the most efficient way to increase the carrier density of graphene and adjust the work function. Substitutional doping of graphene that replaces carbon atoms with hetero atoms (nitrogen, boron) is a possible approach for the stable doping of graphene [12]. This chemical doping is very durable because the dopant forms a covalent bond with a carbon network. However, the substituted atoms cause severe structural damages to the graphene, and ultimately cannot increase the overall conductivity. Therefore, non-covalent doping has been preferred as a non-destructive approach for graphene doping [13]. Various dopants and methods such as spin-coating, dip-coating, and vaporization for doping graphene have been suggested. However, the increase of the carrier density by the non-covalent doping was not sufficient, or the doping stability was also not good enough to be used in practical applications because dopants were physically adsorbed to the graphene. In addition, most of the studies focused on *p*-type doping of graphene, which increases the work function of graphene [14,15,16,17]. On the other hand, since graphene is difficult to maintain in the *n*-type behavior due to moisture and oxygen in the atmospheric environment, research on graphene *n*-type doping that decreases the work function of graphene has been rarely conducted [18,19,20,21]. 

Organic amine is a promising dopant for graphene *n*-doping, because lone-paired electrons in the amine unit can be readily transferred to graphene through charge transfer [22,23]. Polyethyleneimine (PEI) contains a relatively large amount of amine, so more electrons can be transferred to graphene. Kippelen et al. confirmed that the work function of various materials such as metal oxides, metals, poly(3,4-ethylenedioxythiophene) polystyrene sulfonate (PEDOT:PSS) and graphene could be decreased by coating 10 nm thick PEI on each surface [24]. Since then, PEI has been used to decrease the work function of various materials. For example, PEI was used to increase the efficiency of organic light-emitting diodes (OLEDs) by lowering the work function of ZnO, an electron injection layer of OLEDs [25]. In addition, PEI was used to prevent interfacial mixing between interlayer materials and reduce electron injection barriers when fabricating OLEDs by the solution process [26]. Using these advantages, several studies have been conducted to reduce the resistance of graphene by using PEI as an *n*-type dopant. Still, it has not been successful in obtaining a resistance low enough for its practical application [27,28,29].

In this study, PEI, an amine-rich polymer, was adsorbed very thinly on the graphene surface. Then, an *n*-type doping effect was maximized through vacuum annealing, and the work function of the graphene was reduced to 3.66 eV. Through the inter-stacking of the PEI layer and graphene, the resistance of the graphene was decreased to 38.26 ohm/sq while maintaining an optical transmittance of 94.05% at 550 nm. The figure of merit (FoM) of the graphene electrode is as high as 158, which is significantly higher than the minimum standard for commercially available transparent electrodes (FoM = 35) as well as graphene electrodes doped with previously reported chemical doping methods. In addition, the resistance change was less than 25% when the temperature was from 100 K to 500 K in a high vacuum atmosphere, indicating excellent thermal and vacuum stability. Finally, even when immersed in various solvents with high solubility for PEI, the resistance change was within 5%, indicating that the vacuum annealing process enables a strong charge-transfer interaction between PEI and graphene.

## 2. Materials and Methods 

### 2.1. Synthesis of Graphene

A poly-crystal graphene monolayer was grown using a low-pressure chemical vapor deposition (LPCVD) method on a Cu foil (0.025 mm thick, Alfa Aesar, Waltham, MA, USA) Cu foil was immersed in nitric acid in deionized water (DI) for 40 s and then washed in DI to remove the native oxide. The Cu was immediately loaded into an LPCVD chamber and annealed at 1000 °C (heating rate 15 °C/min) for 30 min in a hydrogen atmosphere (100 sccm). Then graphene was grown at 1000 °C, 60 min, and total pressure was 100 torr (CH_4_:H_2_ = 1:100). After growth, as grown graphene on Cu was cooled down to room temperature under vacuum.

### 2.2. Graphene Transfer and Doping Method

Polymethyl methacrylate (PMMA; 2.7 g dissolved in 50 mL chlorobenzene, Sigma Aldrich, St. Louis, MO, USA) was spin-coated (3000 rpm, 60 s) to graphene synthesized on Cu (Gr/Cu) and baked for 30 min on a hot plate. After that, PMMA/Gr/Cu was immersed in 0.1 M ammonium persulfate aqueous solution [(NH_4_)_2_S_2_O_8_] to chemically etch Cu for 1 h. PMMA/Gr was floated on deionized water (DI) to remove residual ammonium persulfate. PMMA/Gr floating on DI was lifted to the various substrates (SiO_2_ (300 nm)/Si, PEI/SiO_2_/Si, Quartz, PET and so on). And then PMMA/Gr/substrate was baked on a hot plate at 100 °C for 1 h. PMMA was removed by dipping in acetone. The process of forming the octadecyltrichlorosilane (OTS) self-assembled monolayer (SAM) on SiO_2_/Si (OTS-SiO_2_/Si) is detailed in our previous reports [23].

Polyethyleneimine (PEI, M_w_: ~25,000, 0.25 g, Sigma Aldrich, St. Louis, MO, USA) was added to 62.8 mL of methanol. The proportion of PEI in methanol was 0.5 wt %. After that, PEI was mixed with methanol through 1 h sonication and 1 h stirring. First, 0.5 wt % PEI in methanol solution was spin-coated on the graphene. Then, it was baked for 10 min at 100 °C to evaporate the methanol completely. After that, vacuum annealing was performed at 70~200 °C for 2 h to maximize the density of free amine in PEI.

### 2.3. Characterization of Graphene

The ratio of free amine and protonated amine of PEI on Gr was analyzed through X-ray photoelectron spectroscopy (XPS, ESCA2000, VG microtech, London, UK). To reduce the influence of the SiO_2_/Si substrate, PEI and Gr layers were formed on the Au deposited SiO_2_/Si substrate. Water contact angle was measured by home-made contact angle measurement system. Water contact angle was also calculated by home-made measurement system. A Raman microscope (WITEC Raman system, Ulm, Germany) with excitation energy of 532 nm was used to analyze the various graphene. The thickness of PEI and the work function of graphene before and after doping were measured by the non-contact mode and kelvin probe force microscope (KPFM) mode of an atomic force microscope (AFM, NX10, Park systems, Suwon, Korea).

### 2.4. Device Fabrication and Characterization

Graphene Field-Effect Transistors (GFETs) were fabricated using a typical photolithographic process. First, a photoresist (PR, AZ 5214E) was coated onto graphene on a substrate at 4000 rpm for 50 s. Then, the graphene channel region was patterned using photolithography. Then, the graphene was removed by oxygen plasma, except for the channel region. After removing the PR, the electrode of the FET was patterned through the second photolithography. Cr/Au (5/40 nm) electrodes were deposited by the thermal evaporator. Then, metal except for electrode region was lifted-off by dipping in acetone. The electrical characteristics of each GFET were measured using a Keithley SCS-4200 system (TEKTRONIX, Oregon, OR, USA) and a vacuum probe station (MST5000, MSTECH, Hwaseong, Korea).

### 2.5. Graphene and PEI Multi-Stacking for FTCEs

To double-side dope the Gr, PEI was spin-coated onto the substrate (SiO_2_/Si or PET or Quartz). Then, it was baked for 10 min at 100 °C to evaporate the solvent (methanol) completely. After that, vacuum annealing was performed at 200 °C for 2 h. Gr was wet-transferred from the anhydrous ethanol solution to the PEI/substrate and baked for 2 h at 100 °C. After removing PMMA, Gr/PEI/substate was annealed at 200 °C for 2 h. Then, PEI was spin-coated on Gr/PEI/substate. Gr was double-side doped (Gr/PEI/Gr) by annealing again for 2 h at 200 °C. The multi-stacked structure was further stacked by repeating the above-described process.

## 3. Results and Discussion

Polyethyleneimine (PEI), an amine-rich macromolecule, can easily transfer electrons to graphene because its work function (3.5~3.7 eV) is lower than that of graphene (4.4~4.6 eV) [30]. As mentioned before, graphene can be *n*-type doped by electron-transfer from lone-pair electrons present in the nitrogen atoms of alkyl-amine [22,23,25,26,27,28]. In previous studies for graphene doping using PEI, PEI diluted with a solvent was coated onto graphene, or PEI itself was vaporized and adsorbed onto graphene [27,28,29]. However, since the p*K*_a_ value of the alkyl-amine is 8 to 11, the amine (–NH_2_) units are easily protonated to the form of alkyl-ammonium (–NH_3_^+^) ions in a normal atmospheric environment [31]. Thus, the doping effect is inevitably reduced. In this study, the protonated amines were effectively transformed into free amines through the vacuum annealing process for maximizing the *n*-type doping effect of graphene (Figure 1).

The deprotonation of amine units on graphene by vacuum annealing was investigated by X-ray photoelectron spectroscopy (XPS) and Raman spectroscopy (Figure 2A,B). The intensity ratio of XPS peaks corresponding to protonated amine (–NH_3_^+^, 400.6 eV) and free amine (–NH_2_, 399.5 eV) indicate that the ratio of free amine units increased from 60% to more than 82% after vacuum annealing (Figure 2A) [25,32]. We note that the ratio of free amine would be higher than 82% since the amine units of the vacuum-annealed sample would have been partially protonated again during sample transfer in air. Also, the water contact angle of the PEI adsorbed graphene surface was changed from 22° to 71° after vacuum annealing (Inset of Figure 2A). The reduced hydrophilicity of vacuum annealed PEI on graphene further indicates that the ratio of polar NH_3_^+^ was significantly reduced. The *n*-doping behavior of graphene by adsorbed amine units can be easily estimated by Raman Spectroscopy because the shift in G and 2D peaks is a signature of graphene doping. Since the G-peak in the Raman spectrum of graphene comes from the single-resonance of graphene with light [33], the G-peaks were shifted in a positive direction due to the *n*-doping of graphene (Figure 2B). In contrast, 2D peaks were shifted in the negative direction because they are formed by double-resonance between K-point and K’-point. The 2D/G ratio and full-width half maximum (FWHM) of the G peak also changed according to the *n*-doping effect after vacuum annealing (Appendix A). 

If the Fermi level of graphene upshifts due to *n*-type doping by the free amine of PEI, the work function to the vacuum level decreases. The work function of HOPG and undoped graphene measured by Kelvin probe force microscopy (KPFM) were 4.61 and 4.39 eV, respectively (Figure 2C). In the case of graphene doped with PEI, the work function decreased to 3.93 eV before annealing, and further decreased to 3.66 eV after vacuum annealing. Because of the ambipolar characteristic of graphene, the resistance varies greatly depending on the degree of doping. Therefore, the resistance is highest at the Dirac-point, and as the carrier density increases by *n*-type doping, the Fermi level upshifts and the resistance decreases. So, when graphene is on hexagonal boron nitride or the self-assembled monolayer (SAM) of octadecyltrichlorosilane (OTS), the Fermi level of graphene is close to the Dirac-point [34,35]. Thus, the resistance of graphene transferred onto OTS-SAM formed on the SiO_2_/Si (OTS-SiO_2_/Si) substrate was as high as 4008 ± 46 ohm/sq through four-point probe measurement (Figure 2D). After n-doping with PEI adoption, the sheet resistance decreased to 332 ± 18 ohm/sq, and significantly further reduced to 224 ± 10 ohm/sq after vacuum annealing.

The change of carrier concentration and resistance due to the vacuum annealing was evaluated by fabricating GFETs (Figure 3). Under high charge density conditions, the resistance of graphene is generally inversely proportional to carrier mobility and carrier density. The Dirac-point voltage of GFETs on the OTS-SiO_2_/Si substrate was close to 0 V, and sheet resistance of the device at zero gate vias was above 4000 ohm/sq, similar to the result of the four-point probe measurement (Figure 3A). When graphene was *n*-type doped using PEI, the Dirac-point was shifted to −205 V and the sheet resistance decreased to 335 ohm/sq. When the free amine density on graphene was maximized through vacuum annealing, the Dirac-point shifted to a much more negative voltage than −210 V (lowest measurable voltage) and the resistance significantly further decreased to 229 ohm/sq. The temperature for vacuum annealing to maximize the doping effect was 200 °C. Since 200 °C is a relatively high temperature to be applied to organic optoelectronic devices such as OLED, vacuum annealing at a low temperature (70 °C, 2 h) was also conducted (Appendix A). As a result, the difference in sheet resistance between the graphene devices annealed at 70 °C and 200 °C was as small as 46 ohm/sq. This result proves that the doping method we propose can fully maximize the doping effect even with a low temperature vacuum annealing process, and thus can be incorporated into an organic optoelectronic device. Since graphene is a mechanically stable one-atom-thick layer, both sides of the graphene layer can be simultaneously doped. We confirmed that, when dual sides of graphene were doped with PEI, the doping effect was enhanced, and the sheet resistance was further decreased to 121 ohm/sq. In order to quantitatively calculate the increased electron density of graphene through doping, it is necessary to know how much the Dirac-point of the GFETs has shifted. However, in the case of heavy *n*-doped graphene, Dirac-point and electron concentration were estimated through simulation using a Drude model and a parallel capacitor model because the Dirac-point does not exist within the measurement range [36]. Compared to carrier concentration (8.3 × 10^11^ cm^−2^) before annealing, in the case of one-side doping after annealing, the Dirac-point shifted to −310 V, and the carrier density increased to 3.9 × 10^12^ cm^−2^. After the dual-side doping, the Dirac-point was shifted up to −768 V, and the carrier density was increased to 2.0 × 10^13^ cm^−2^. Before doping, since the Dirac-point of GFETs was 1 V and the carrier density was 3.6 × 10^11^ cm^−2^, the PEI doping method that maximized free amine through vacuum annealing proved to be a heavily effective doping method. 

The doping effect can be controlled by the applied voltage. Due to the 40% protonated amine present in PEI doped GFETs before annealing, the doping tendency varied depending on the applied gate voltage. As a result, the n-doping effect of the PEI doped GFET increased and the Dirac-point negatively shifted according to the negative potential increase from −80 V to +80 V and −150 V, −180 V, −210 V to +80 V (Appendix A). On the other hand, most of the amines of PEI-doped GFETs exist as free amine and are already heavy *n*-doped and are thus hardly affected by the applied voltage after annealing (Appendix A).

For the FTCE application of graphene, high optical transmittance is an important requirement, as well as low sheet resistance. The optical transmittance of monolayer graphene used in this work was 97.3 ± 0.3% at a wavelength of 550 nm, which is close to the theoretical value of 97.7%. PEI doped graphene showed the transmittance of 97.0 ± 0.5%, which indicates PEI reduced the transmittance by only 0.3%. The reason that the PEI layer has little effect on the transmittance is that the layer is very thin (~3 nm) and is evenly formed on the graphene surface (Appendix A). These results indicate that, through double-sided doping and additional multiple stacking of graphene (G) and PEI (P), the sheet resistance of the graphene electrode can be easily reduced without a significant decrease in its transmittance. When P/G/P structures were fabricated by additional PEI doping on graphene/PEI/SiO_2_/Si (G/P) structures, the sheet resistance was reduced to 118 ± 15 ohm/sq at 96.7% transmittance (Figure 3B). When graphene was additionally stacked to form a G/P/G/P structure, the sheet resistance decreased to 67.6 ± 1.7 ohm/sq at a transmittance of 94.3%, and finally, when a P/G/P/G/P structure was formed, the sheet resistance was dramatically reduced to 38.3 ± 1.5 ohm/sq at the transmittance of 94.0%. The FoM of a P/G/P/G/P electrode, which was converted from its transmittance and sheet resistance, is ≈158 (orange 14 in Figure 3C). This FoM is much higher than previously reported *n*-doped graphene (Group I) and *p*-doped graphene (Group II) (Figure 3C), showing that the method proposed in this study is very effective for heavy *n*-type doping of graphene.

The additional advantage of the graphene doping using PEI is the excellent chemical and thermal stability of its dopant effect. The branched PEI used in this study can be strongly attached to the graphene surface because of its large molecular size and the strong charge-transfer interaction between nitrogen (N) atoms of PEI and graphene. To evaluate the thermal stability of the *n*-doping effect by PEI, the resistance of PEI/Gr/SiO_2_(300 nm)/Si structure was measured at various temperatures under a vacuum atmosphere (~10^−5^ torr) (Figure 4A). The temperature was stabilized for 20 min for each temperature before measurement. After vacuum annealing, when the resistance measured at 300 K was normalized to 1, when the temperature was reduced to 100 K, the resistance decreased to 0.89 ± 0.11, due to a decrease in phonon scattering and an increase in carrier mobility [37,38]. When the temperature was raised to 500 K, the resistances increased to 1.22 ± 0.17 because of increased phonon scattering. When the device was measured again at room temperature, the sheet resistance increased by only 2.7% compared to the initial value of 300 K. This result indicates that the PEI used as a dopant is not detached in a high vacuum or low/high temperature environments.

The chemical stability of PEI was also confirmed by dipping the PEI doped graphene in various solutions. First, after dipping in a solvent with high PEI solubility for 30 min, the resistance changes of the PEI/Gr/SiO_2_/Si electrode were measured. Solvents were classified and selected into three types: polar protic, polar aprotic and non-polar solvent. Despite being immersed in methoxyethanol, methanol, chloroform and dimethyl sulfoxide (DMSO) with high PEI solubility for 30 min, the overall resistance change was less than 2.1%. This result proved that the charge transfer interaction between graphene and PEI is robust, especially after the vacuum annealing. The change in the doping effect in aqueous solutions of various pH was also investigated by dipping GFETs in various pH solutions (0.1 M) for 30 s. As expected from p*K*_a_ (9.06) of the amine of PEI [39], when the pH is lower than 9, some amines are protonated, reducing the doping effect and increasing resistance (Figure 4C). However, in strong basic solutions (NaOH) above pH 9, the resistance was further reduced by 1.4%. This is probably because the small amount of protonated amine remaining after annealing was further deprotonated to free amine. Since the doping effect of graphene depends on the pH of the solution, PEI was not separated from the graphene, indicating that the PEI doping is chemically stable.

Amine is protonated by moisture or oxygen in the atmosphere to become an ammonium ion (−NH_3_^+^). Therefore, the sheet resistance of PEI-doped GFETs was increased because the doping effect was decreased when exposed to air (Appendix A). Sodium metasilicate (Na_2_SiO_3_) was formed on the PEI as a passivation layer, and PEI was prevented from being directly exposed to air. As a result, despite being exposed to air for 14 days, the resistance increased by only 25%. In the case where the passivation layer was not formed on the device (PEI/Gr/SiO_2_/Si), the sheet resistance increased by about 300 times after 14 days, but the sheet resistance could be lowered almost similarly to the resistance measured in vacuum by 150 °C, 2 h annealing. 

As an FTCEs material, flexibility is as important as sheet resistance and permeability. Since both PEI and Gr are flexible, we assume that they are suitable as FTCE materials. Therefore, in order to verify the flexibility of PEI-doped Gr, a bending test of a Gr/PEI/PET (75 µm) structure was performed. As a result, the sheet resistance increased only 2.7% after 5000 bendings. Based on these results, it was verified that the PEI-doped Gr proposed in this study can be sufficiently used as an FTCE material.

## 4. Conclusions

In conclusion, we demonstrated the strong *n*-doping effect of the amine-rich macromolecule on graphene. By maximizing the free amine density through vacuum annealing, the carrier density of graphene was dramatically increased, and the sheet resistance of graphene was significantly decreased to 38 ohm/sq by charge transfer doping. In addition, we were able to fabricate FTCEs with a high figure of merit (FoM ≈ 158) because the dopant layer is very thin and transparent. Besides, the *n*-type doped graphene using PEI not only showed excellent flexibility and chemical stability, but also maintained the doping effect even in high temperature and high vacuum conditions. These results indicate that maximizing the *n*-type doping effect on graphene through vacuum annealing is effective in obtaining stable and very conductive graphene FTCEs. Furthermore, our approach can be applied to the surface-doping of other two-dimensional materials without degrading their electrical properties.

## Figures and Tables

**Figure 1 materials-13-02166-f001:**
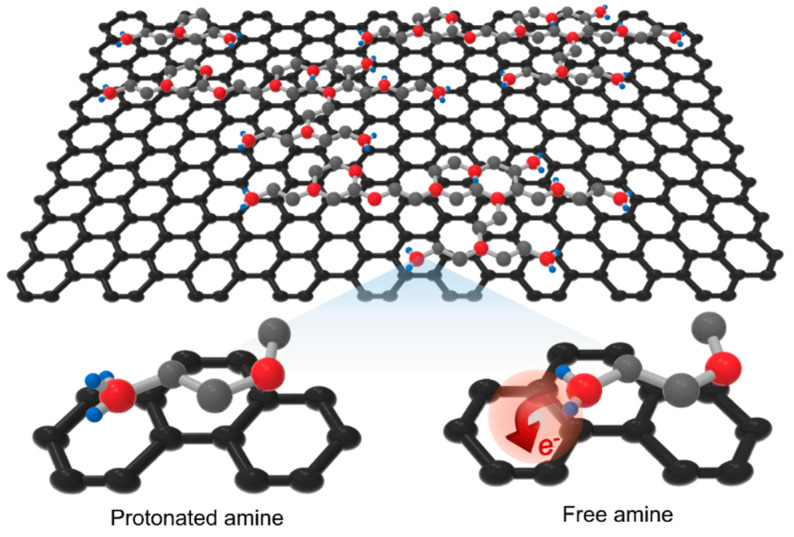
Schematic image of PEI doped graphene and molecular structures of protonated and free amine.

**Figure 2 materials-13-02166-f002:**
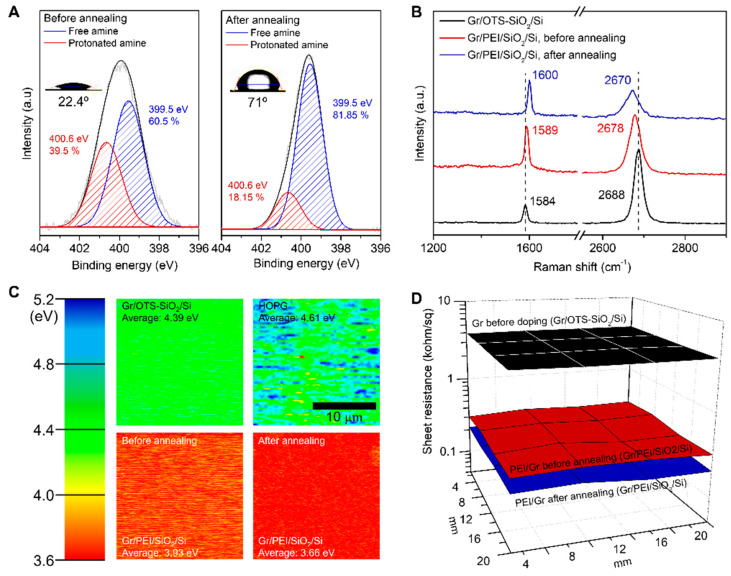
Characteristics of *n*-type doped graphene using PEI before and after vacuum annealing. (**A**) Comparison of protonated amine and free amine ratio. The left side graph is PEI/Gr/Au/SiO_2_/Si before annealing, and the right side graph is after annealing analyzed by XPS. (**B**) Raman spectra of graphene before/after doping & vacuum annealing. (**C**) Work function of each graphene. All graphene was measured by Kelvin probe force microscope (KPFM). Highly oriented pyrolytic graphite (HOPG) is a reference to confirm that the work function of graphene analyzed by KPFM is correct. (**D**) Sheet resistance uniformity (20 × 20 mm^2^) of PEI doped graphene was analyzed by four-point probe measurement. Average sheet resistance of graphene before doping (Gr/OTS-SiO_2_/Si) was 4007 ± 46 ohm/sq, Gr/PEI/SiO_2_/Si before annealing was 332 ± 18 ohm/sq, and after annealing was 224 ± 10 ohm/sq.

**Figure 3 materials-13-02166-f003:**
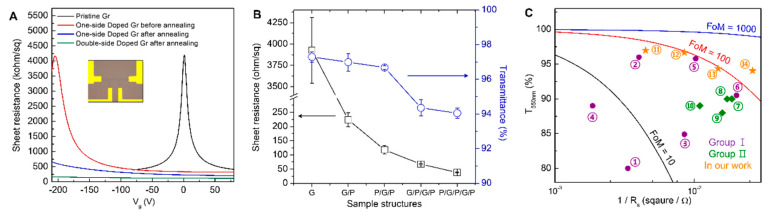
Characteristics of transparent conducting electrodes, such as sheet resistance and transmittance. (**A**) Gate-dependent sheet resistance of graphene before/after doping and vacuum annealing. The black line consists of Gr/OTS-SiO_2_/Si, red and blue lines are Gr/PEI/SiO_2_/Si, and the green line is PEI/Gr/PEI/SiO_2_/Si. Inset is a photo image of GFETs. Channel length and width are 20 µm and 5 µm, respectively. (**B**) The transmittances and sheet resistance change of multi-stacked graphene (G) and PEI (P). (**C**) Figure of merit (FoM) for transparent conducting electrodes. Purple circles (Group I, ①~⑥ are reference 18~23 of this papers) are previous reported graphene *n*-doped using amine or another dopant, green diamonds (Group II, ⑦~⑩ are reference 14~17 of this papers) are previous reported *p*-doped graphene, and orange stars (⑪~⑭) are the *n*-doped graphene in this study. FoM was calculated by following the equation in Ref [5]. σ_dc_/σ_opt_ = 188.5/R_sheet_ (T_550nm_^−1/2^ − 1), where σ_dc_ and σ_opt_ represent direct current conductivity and optical conductivity at a wavelength of 550 nm, respectively.

**Figure 4 materials-13-02166-f004:**
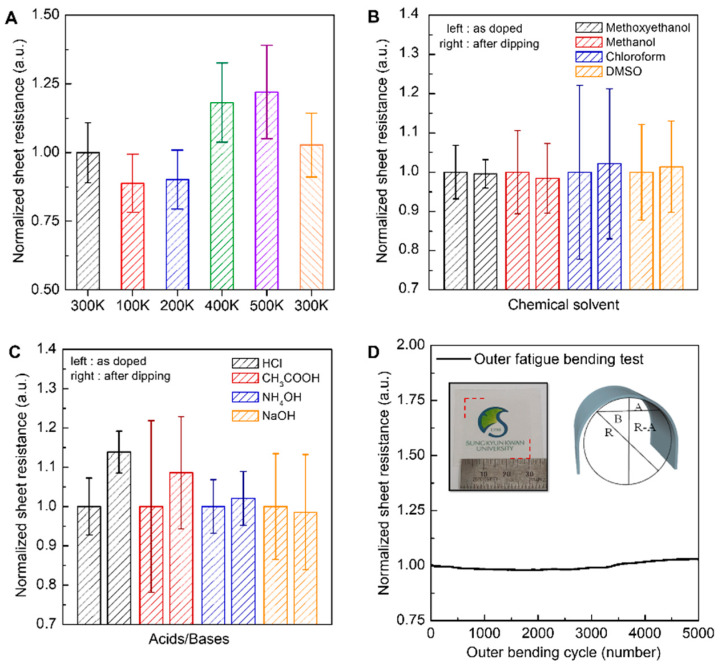
Chemical and thermal stability and flexibility of the graphene electrode *n*-doped using PEI (PEI/Gr/SiO_2_/Si). (**A**) Change of sheet resistance at high vacuum (5 × 10^−6^ torr) according to temperature change. (**B**) Resistance change after dipping in various organic solvents for 30 min. (**C**) Resistance change after dipping in water with various pH for 30 s. (**D**) Outer bending stability test of Gr/PEI/PET (75 µm). The radius range is 15 to 5 mm. The left inset image is the photo image of Gr/PEI/PET. The bending radius was calculated using the formula B^2^ + (R − A)^2^ = R^2^, as shown in the right inset of Figure 4D).

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
