# Peer review of "Highly Efficient n-Type Doping of Graphene by Vacuum Annealed Amine-Rich Macromolecules"

_materials, 2020, doi:10.3390/ma13092166_

Round 1
Reviewer 1 Report
In this paper, the author(s) reported the strong n-doping effect of the amine-rich macromolecule on graphene, and demonstrated that the n-doped graphene electrode shows not only excellent chemical stability, but also thermal stability. The study of this n-doping is very interesting, although some expression should be polished. Once these points are corrected, the manuscript is ready to be published in the Materials.
- Figure 1D on Page 4 (line 140) cannot be found in the manuscript. And all of the figures in the manuscript are blurred, please re-confirm and upload the clearer figures of the experiment results.
- The author(s) claims that the n-doped graphene electrode can be used as the flexible transparent conducting electrode (FTCE). However, the substrates used in the manuscript were SiO2/Si, and there were none experimental results related to the flexible characteristics of the electrodes. The author(s) should make a clear assessment of the flexibility of this electrode.
- Different layers of graphene will seriously affect the sheet resistance and transmittance of graphene electrodes. Regarding the electrode structures of G/P/G/P and P/G/P/G/P mentioned in the manuscript, the author(s) need to further explain whether the improvement of photoelectric performance is only due to the increase of graphene layers. Moreover, the improvement in performance brought by the stacking of the layers cannot explain the superiority of doping. Please add the experiments and provide performance of the undoped bilayer graphene film (G/G structure) for comparison.
- The surface roughness is also one of the important parameters for evaluating transparent electrodes. The doping process reduced the hydrophilicity, which must increase the difficulty of the transfer process of graphene. Therefore, the author(s) need to illustrate the effect of the doping process and the stacking process on the surface roughness.
- Please provide the duration that the sample was kept at different temperatures in the thermal stability experiment.
- There are many errors in the For example, the format of the unit (ohm/sq or ohm∙cm-2) needs to be unified. The abbreviation of graphene (G or Gr) also needs to be unified. Please double check the whole manuscript and modify carefully.

Author Response
1.Figure 1D on Page 4 (line 140) cannot be found in the manuscript. And all of the figures in the manuscript are blurred, please reconfirm and upload the clearer figures of the experiment results.
Response 1
- Figure 1D is a typo.
- Modified to Figure 2C.
2. The author(s) claims that the n-doped graphene electrode can be used as the flexible transparent conducting electrode (FTCE). However, the substrates used in the manuscript were SiO2/Si, and there were none experimental results related to the flexible characteristics of the electrodes. The author(s) should make a clear assessment of the flexibility of this electrode.
Response 2
- Since PEI and Gr are both flexible materials, their properties will not change even after multiple bending tests.
- To verify this, a flexibility test of PEI-doped Gr was conducted (Figure 4D).
- The resistance increased only 2.7% even after 5000 times bending the device of Gr/PEI/PET structure.
- This result proved that PEI-doped Gr could be sufficiently used as an FTCE material.
- Details were added to the 291 line and figure 4D caption.
3. Different layers of graphene will seriously affect the sheet resistance and transmittance of graphene electrodes. Regarding the electrode structures of G/P/G/P and P/G/P/G/P mentioned in the manuscript, the author(s) need to further explain whether the improvement of photoelectric performance is only due to the increase of graphene layers. Moreover, the improvement in performance brought by the stacking of the layers cannot explain the superiority of doping. Please add the experiments and provide performance of the undoped bilayer graphene film (G/G structure) for comparison.
Response 3
- The sheet resistance of double-side doped Gr was extraordinarily lower than one-side doped (Figure 3A).
- In addition, it was mentioned in line 226 ~ 231 that the sheet resistance could be further reduced by double-side doping and additional graphene stacking.
- Lowering the sheet resistance of graphene through layer-by-layer stacking was not mentioned in the main-text because it is a well-known fact, not a newly discovered result in this study.
- Therefore, sheet resistance of P/G/P/G/P surely lower than G/G.
- In addition, since PEI only causes a 0.3% decrease in transmittance, it is very suitable to fabricate a multi-stacked structure.
- It is fact that the resistance difference occurs when only graphene is stacked and when there is a dopant between Gr and Gr, it is described in detail at https://doi.org/10.1021/nn1008808.
4. The surface roughness is also one of the important parameters for evaluating transparent electrodes. The doping process reduced the hydrophilicity, which must increase the difficulty of the transfer process of graphene. Therefore, the author(s) need to illustrate the effect of the doping process and the stacking process on the surface roughness.
Response 4
- PEI is simply spin-coated on graphene.
- Therefore, it is judged that no additional illustration is necessary.
- In order to wet-transfer Gr to a hydrophobic surface (OTS, annealed PEI), a transfer process was performed in anhydrous ethanol.
- The detailed stacking process was described in Materials and Methods.
5. Please provide the duration that the sample was kept at different temperatures in the thermal stability experiment.
Response 5
- For each temperature, the temperature was stabilized for 20 min and then measured.
- For that part, we added the content to line 254.
6. There are many errors in the manuscript. For example, the format of the unit (ohm/sq or ohm∙cm-2) needs to be unified. The abbreviation of graphene (G or Gr) also needs to be unified. Please double check the whole manuscript and modify carefully.
Response 6
- Manuscript has been reviewed several times to eliminate typos.
Reviewer 2 Report
Please see attached file.

Author Response
Major revisions:
1) Raman spectra confirms good quality of pristine Gr after transfer on the substrate. Anyway, Gr report on substrate is not described and the impact of Cu residues should be discussed.
Response 1
- There seems to be a misunderstanding because we haven't clarified how we analyzed the properties of pristine Gr and doped Gr
- We chemically etched away Cu of Gr/Cu and transferred Gr to OTS treated SiO2(300nm)/Si
- Then the Raman properties of Gr/OTS-SiO2/Si were analyzed.
- Therefore, Cu residues do not exist.
- Figure 2 has been modified to avoid misunderstanding.
- And the Gr transfer and characterization methods such as XPS, RAMAN, and AFM were added to 2.2 Materials and Method
2) Spin coating of PEI seems already an efficient way to induce n doping of Gr (R decreases from 4 kohm/sq to ~332 Ohm/sq). Low temperature annealing at 70°C (more appropriate for OLED application) increases further the doping (R decreases from 332 Ohm/sq to ~275 Ohm/sq). Is the gain obtained with annealing valuable for application? Can author compare the Dirac point and charge density of PEI spin coated samples with and without annealing?
Response 2
- Compared to before annealing, the sheet resistance decreased about 18% after vacuum annealing at 70℃ 2 hours, so, it is important to increase the doping effect through vacuum annealing.
- In addition, comparing the red line and blue line in Figure 3A, it could be confirmed the difference of the Dirac-points depends on the with and without vacuum annealing.
- Although mentioned in the main text, the carrier concentration increased to 3.9 x 1012 cm-2 after vacuum annealing, which could be calculated by simulation using the Drude model. The carrier concentration before vacuum annealing was 8.3 x 1011 cm-2, which was added to the line 208.
3) n-type doping approaches are unstable in atmospheric conditions due to electron withdrawing by adsorbed charged impurities, such as oxygen and water molecules. After annealing, samples should never see atmosphere to avoid a reduction of the doping effect. Authors should discuss solution to allow for manipulation of modified Gr electrodes in ambient conditions.
Response 3
- Air-stability also main issue of n-doped graphene based TCEs.
- The air-stability test of PEI doped GFETs was conducted, and the content was added to line 283 and figure S5.
- Using sodium metasilicate as a passivation layer, air-stability of PEI-doped GFETs could be improved.
- Also, the sheet resistance could be recovered similarly to the initial measured value through vacuum annealing at 150℃, for
4) Gr is not directly transferred on Si/SiO2 substrate but on OTS-SAM formed on SiO2/Si substrate. Is it the case for all explored samples? Can the doping procedure be applied in an efficient way also in absence of SAM?
Response 4
- When graphene is transferred onto SiO2/Si, it is slightly p-doped by surface charge.
- In order to prevent unwanted doping effect, OTS-SAM formed on SiO2/Si was used as the substrate for confirming pristine Gr characteristics.
- Experimentally, Gr/OTS-SiO2/Si showed the characteristics similar to pristine (Dirac-point of Gr/OTS-SiO2/Si in figure 3A was 1V).
- PEI dopants could be easily spin-coated between Gr and SiO2/Si and on Gr surface regardless of OTS.
5) Authors assert “PEI can be strongly attached to the graphene surface because of its large molecular size and the strong charge-transfer interaction between nitrogen (N) atoms of PEI and graphene”. It could be useful to support such statement by adding a short description of the doping mechanism. Ideally density-functional theory (DFT) calculations of PEI-doped graphene can reinforce this part.
Response 5
- Amine compounds as well as PEI are known to transfer electrons and reduce work function to various materials, including graphene, metal oxide, metal, and polymer (10.1126/science.1218829).
- And the references related to n-doped graphene by charge transfer electron mechanism (from 22 to 29) were already in the main text, additional DFT calculation is unnecessary.
6) Authors should discuss the origin of WF reduction with doping (electronic dipole interaction, interface dipole formation,...).
Response 6
- As in number 5, references (22 to 29) have been in the main-text, so we think further discussion is unnecessary.
7) Can author exclude that external environment, such as high applied electric field for gating and incident light excitation has no effect on surface charge transfer?
Response 7
- PEI is an insulating material and is not excited by light in a general measurement environment.
- Experimentally, the PEI-doped GFET was further doped by a high electric field before vacuum annealing.
- As a result, Dirac-point and sheet resistance PEI-doped GFET differed according to the applied voltage (Figure S4A, B).
- On the other hand, after annealing, the doping effect hardly depended on the applied voltage (Figure S4C).
- We added more information in line 214.
8) Authors should discuss the homogeneity of the doping, the doping efficiency as function of the coverage and the reproducibility of the doping process. Fig S2 shows holes and hills of micrometric sizes indicating a non-uniform coverage, but this point is not discussed.
9) Smooth and uniform surface are required for FTCE applications. Fig. S2 show single AFM lines profiles that poorly allow to evaluate roughness, an RMS value over representative surfaces should be calculated. The quality of the PEI/Gr surface and the effect of roughness should be discussed.
Response 8 and 9
- In this study, graphene was transferred to a substrate through traditional wet transfer method by Cu chemically etching using PMMA as a supporting layer.
- When graphene is transferred by traditional method, Polymethyl methacrylate (PMMA) residues remain on the surface on graphene or wrinkles or tears of graphene occur.
- The holes in Figure S2B were tearing regions of graphene.
- The hills were PMMA residues that remain on the graphene surface or particles that are thousands to hundreds of nanometers thick on SiO2/Si substrates.
- As evidence, Gr surface was non-uniform and the RMS of Gr before PEI coating was 2.34 nm (Newly added as figure S2C).
- On the other hand, the RMS of PEI/Gr was 0.75 nm after PEI coating, indicating that PEI formed relatively uniform film (Fig. S2B, newly added value of RMS).
- In addition, the sheet resistance of 20 x 20 mm2 PEI doped graphene was uniform to 94.7% before annealing and 95.7% after annealing (Figure 2D, detailed values added to caption).
- These results demonstrated that PEI was homogeneous doping of Graphene.
10) Generally, the extent of doping of the graphene depends on the surface coverage and on the structure configuration of the functional group. Can the author discuss the control over doping by the density of the surface dopants?
Response 10
- The purpose of this study is to maximize the doping effect for minimizing the sheet resistance.
- Doping effect could be controlled by adjusting the annealing temperature as shown in figure S3 even though it is difficult to control the density Since the dopant is spin-coated on the graphene surface.
- In addition, as shown in figure 4C, the free amine density could be controlled by dipping in acid or basic solution.
11) What is the stability over time? Have authors verified long-term stability of the doping (i.e. by measuring the samples again after one month?
Response 11
- The result of air-stability was added to figure S5.
- The sheet resistance of PEI/Gr/SiO2/Si structure GFETs with sodium metasilicate passivated increased by 25% after 14 days.
- However, sheet resistance is expected to increase to less than 30% even after one month because sheet resistance has hardly increased since 72h.
- In addition, the increased sheet resistance is reduced again to the initial value after vacuum annealing at 150℃ for 2 hours.
- Through these results, long term stability was sufficiently demonstrated.
12) Have multi-stacked Gr/PEI samples analysed in Fig.3 been subjected to the annealing procedure? If yes, is there any difference with non-annealed multi-stacked samples
Response 12
- All multi-stacked Gr/PEI samples were annealed at 200℃ for 2 hours.
- The multi-stacked Gr/PEI samples also had differences in sheet resistance before and after annealing, but the differences were not significant.
- This is because graphene has already been sufficiently doped.
- Therefore, we think that experimental results before and after annealing multi-stacked Gr/PEI samples are unnecessary.
13) Surface modified Gr based FTCE electrodes are the focus of the Gr doping method proposed. Why bending tests have not been performed? Authors should enrich their work by presenting also performance of the realized device under bending.
Response 13
- Since PEI and Gr are both flexible materials, their properties will not change even after multiple bending tests.
- To verify this, a flexibility test of PEI-doped Gr was conducted (Figure 4D).
- The resistance increased only 2.7% even after 5000 times bending the device of Gr/PEI/PET structure.
- This result proved that PEI-doped Gr could be sufficiently used as an FTCE material.
- Details were added to the line 291 and figure 4D caption.
Minor points:
14) It can be useful to have a device image and information about dimensions.
Response 14: Device image was added to Figure 3A, and the channel length and width were specified in the caption.
15) Details on how contact angle is measured (model for surface energy calculation, reagent used) can be useful in SI for example.
Response 15: Content related to contact angle was added to Materials and Method.
16) Authors should give an estimation of the mobility of the modified Gr, which is also a useful parameter to characterize FTCE.
Response 16
- The PEI doped Gr has a much higher carrier concentration than the pristine Gr, which means that the mean free path of Gr is reduced and consequently the mobility is reduced.
- Mobility of extracted by simulation Gr/OTS-SiO2/Si was 4377 cm2/V∙s, Gr/PEI/SiO2/Si before annealing was 1812 cm2/V∙s, Gr/PEI/SiO2/Si after annealing was 1244 cm2/V∙s, PEI/Gr/PEI/SiO2/Si before annealing was 873 cm2/V∙s
- However, this mobility value is approximate value from simulated Dirac-point and graph shapes, so it is unreliable and misleading.
- Therefore, I think it is better not to suggest this mobility.
17) Units in AFM profile of Fig. S2 have been inverted between x and y axis, I guess.
Response 17: We modified x and y in Figure S2.
18) On line 148 and line 163, unusual comma is used for thousand indication. This can be misunderstanding.
Response 18: Comma was removed.
19) Revisions of the English language can be useful in some part of the manuscript.
Response 19: Manuscript was reviewed several times to correct English.
Round 2
Reviewer 2 Report
Authors have extensively answer to the asked questions and improved the quality of the manuscript.
The paper can be accepted for publication in Materials journal in the present form.
Best regards